# R&D Investment, Financial and Environmental Performance Nexuses via Bootstrap Fourier Quantiles Granger Causality Test

Feng-Li Lin 

Department of Accounting, Chaoyang University of Technology, Taichung 41349, Taiwan; bonnie@gm.cyut.edu.tw

**Abstract:** This study investigated the relationship between R&D investments and financial and environmental performance. The direction, size, and significance of various phases of these variables were generated using the bootstrap Fourier quantiles Granger causality test. In our results, a positive relationship between R&D investment and $CO_2$ emission reductions was found at two tails of quantiles. Additionally, we observed a significantly positive relationship between financial performance and $CO_2$ emission reductions at the 0.5 quantile and above. The correlation between R&D investment and financial performance was identified to be positive under the 0.3, 0.4, 0.5 and 0.9 quantiles and negative under the 0.5 and 0.6 quantiles. The changing linkages among R&D investment, environmental performance and financial performance found in this study provide important information for policy makers, aiding in the development of R&D strategies to upgrade financial and environmental performance simultaneously.

**Keywords:** environmental performance; financial performance; R&D; bootstrap Fourier quantiles Granger causality test



## 1. Introduction

When considering the influence of $CO_2$ emissions on global warming, it is usually accepted that firms play a key role in emitting high levels of air contaminants through the manufacturing process (Busch and Hoffmann 2011). Widespread social concerns about this issue have pushed firms to find out ways of reducing their $CO_2$ emissions. Nevertheless, firms have encountered ostensible dilemmas relating to potential advantages and disadvantages. On the one hand, decreasing $CO_2$ emissions may not only create a cost burden for firms, but may also be harmful to competitiveness; on the other hand, it may save costs and improve efficiency (Hart and Ahuja 1996). Thus, the basic issue currently facing firms is how to develop strategies to upgrade firms' environmental performance (FEP) as well as financial performance (FFP) simultaneously.

Moreover, innovation is a strategy, i.e., several firms have lowered $CO_2$ emissions, upgraded competitiveness and improved FFP concurrently (Triguero et al. 2013). For this reason, this innovation is a key outcome of firms' research and development (R&D) investments; R&D investment has a positive influence on FFP simultaneously (Scarpellini et al. 2019; Duque-Grisales et al. 2020; Ezhilarasi and Kabra 2020). Relevant green R&D investment not only reduces $CO_2$ emissions, upgrading FEP (Samsul et al. 2019), it also improves FFP (Lee and Min 2015) and firm value (Ganda 2018). In summary, previous studies conclude that the greater the R&D investment, the greater the $CO_2$ emission reduction (better FEP). However, Duque-Grisales et al. (2020) state that green R&D investment is not associated with better FFP.

Additionally, prior studies have focused on how FEP responds to FFP, using $CO_2$ emissions as a proxy for FEP with respect to corporate social responsibility. Several studies have revealed the negative nexus between $CO_2$ emissions (FEP) and FFP (Liu et al. 2017; Moneva and Ortas 2010; Clarkson et al. 2011; Gallego-Álvarez et al. 2015), especially

during the pre-financial crisis period (Muhammad et al. 2015). Other studies have revealed negative associations between $CO_2$ emission and firm value (Saka and Oshika 2014; Matsumura et al. 2013). However, several studies have disclosed a negative nexus between FEP and FFP (Liang and Liu 2017) or firm value (Chang 2015). Others have found no such relationships (Lucato et al. 2017), including during the financial crisis (Muhammad et al. 2015). In addition, some have investigated the causal nexus between FEP and FFP; Endrikat et al. (2014) have shown not only that this causal direction might be reversed, i.e., from FFP to FEP, but also that this nexus is positive and partially bidirectional. Hence, to date, relevant studies of the relationship between FEP and FFP have not reached consistent conclusions.

This study examined the nexus between R&D investment, FEP ($CO_2$ emissions) and FFP in Taiwan using a novel bootstrap Fourier quantiles Granger causality test (BFQGCT) and data from 2005 to 2018. To date, this BFQGCT study is the first to provide a deep analysis of the influence of direction and size, as well as their significance in various phases of these trivariate relations. The empirical results show that the greater the R&D investment, the greater the $CO_2$ emission reduction under 0.2, 0.3, 0.7, 0.8 and 0.9 quantiles, separately; meanwhile, the greater the reduction in $CO_2$ emissions, the better the FFP under 0.5, 0.6, 0.8 and 0.9 quantiles; the more R&D investment, the better the FFP under 0.3, 0.4, 0.5 and 0.9 quantiles, respectively; moreover, the better the FFP, the less R&D investment under 0.5 and 0.6 quantiles, individually. Hence, understanding the causality nexus, as well as pathways among R&D investment, FEP and FFP, may have special significance for policy makers, helping to understand how R&D strategies can upgrade FEP and FFP concurrently.

This study is organized as follows: The next section discusses the literature review. The subsequent section proposes the new bootstrap Fourier quantiles Granger causality test. We then describe the data used in the study; then, the empirical results are offered. The last section contains the proposals and conclusions of this study.

## 2. Literature Review

Despite the huge number of studies regarding the R&D investment–FEP–FFP nexus for various countries over various periods, varying outcomes continue to be observed and the whole picture remains unclear. The trivariable nexus is conventionally analyzed in three parts: R&D investment–FEP, R&D investment–FFP and FEP–FFP. For the first strand of the literature, Lee and Min (2015) indicate the influence of green R&D investment in eco-innovation with regard to FEP and FFP in Japanese manufacturing between 2001 and 2010, finding a negative nexus between $CO_2$ emissions and green R&D, although the latter was positively associated with FFP. Using firm-level data from G6 countries between 2004 and 2016, Samsul et al. (2019) showed that R&D investment upgrades FEP. Utilizing data from 181 technology-industry firms in Taiwan, Huang and Wu (2010) reveal that FEP in green commodity initiatives displayed a positive influence on FFP. Examining the nexus between FFP and green initiatives, as well as FEP, Chen et al. (2018) found that green initiatives displayed a positive effect upon green FEP, which in turn exerted a positive influence on FFP. Nevertheless, the influence of green initiatives varies by nation. Firms in Canada, as well as European nations, have linked green initiatives and FEP, as well as in Japan and USA. Hong Kong and China lag behind compared to other nations. Tariq et al. (2019) found that the higher the green commodity innovation performance, the higher the FFP in Thai manufacturing firms.

Several pieces of empirical evidence of a nexus between R&D investment and FFP exist. Lin (2017) found that R&D investment has a positive link with a firm's value in a study of firms engaging in social responsibility in Taiwan. Ganda (2018) reported a positive nexus between Green R&D and firm value in an analysis of 14 South African mining companies. Analyzing green patents and R&D intensity, with the aim of discovering the factors leading to successful eco-innovation procedures in Spain and in the EU, Scarpellini et al. (2019) proposed a positive nexus between eco-innovation activity and R&D intensity in firms. Meanwhile, relevant results show the impact of implementing

innovation on FFP. Ezhilarasi and Kabra (2020) found that R&D investment has a positive influence on FFP. Duque-Grisales et al. (2020) reported that a positive moderating effect of R&D investment on the nexus between FFP and green innovation was exhibited in firms with increasing R&D investments, analyzing 86 listed firms in emerging transnational markets in Latin America.

Some studies have specified whether it pays to be green, thus concentrating on the nexus between FEP and FFP. Clarkson et al. (2011) addressed the question "Does it pay to be green?" among the four most polluting industries in the US, showing that firms which upgraded their FEP in the previous period resulted in improving their FFP in the following periods. Busch and Hoffmann (2011) reported that when using $CO_2$ emissions as an outcome-based approach, FEP pays off. Contrarily, using $CO_2$ control according to a procedure-based method, they observed a negative nexus between the FEP and FFP among the 2500 biggest firms by market capitalization in the Dow Jones Global Index. Matsumura et al. (2013) indicated that on average, for each extra million kilograms of $CO_2$ emissions, company value is reduced by USD 212,000 for S&P 500 corporations. A negative nexus between $CO_2$ emissions and FFP was found to exist in the UK (Liu et al. 2017) for 89 international firms (Gallego-Álvarez et al. 2015) during the pre-financial crisis period (Muhammad et al. 2015). Molina-Azorín et al. (2009) reviewed 32 studies to examine the influence of FEP on FFP. The results were varied, but studies in which a positive impact of FEP on FFP was found were dominant. Buysse and Verbeke (2003) reported that an environmental leadership approach is linked with an increase in FFP for Belgium-based companies, but not for affiliates of foreign multinational enterprises.

Several studies have evaluated the nexus between FEP and environmental disclosure, studying whether investors consider the significance and value of FEP information when making investment decisions. Dilla et al. (2019) showed that FEP, as well as assurances regarding relevant FEP information, had a high effect on investors' investing decisions, with sturdy environmental responsibility opinions. DiSegni et al. (2015) compared the FFP in all the US companies included in the Dow Jones Sustainability Indices and proposed that corporations with strong social responsibility and environmental sustainability are categorized by importantly larger profit measures than the corresponding industries and firms. Wingard and Vorster (2001) reported a positive nexus between FFP and firm environmental responsibility with respect to South African recorded firms, i.e., the higher the level of a firm's environmental responsibility, the better its FFP. Longoni and Cagliano (2018) offered empirical support for the influence of comprehensive environmental disclosure practices in relation to FFP but found no nexus with respect to the effect on FEP. Nevertheless, Deswanto and Siregar (2018) showed that FFP has no effect on firm environmental disclosures. Moreover, they argued that lagging FEP indicates a positive influence with respect to firm environmental disclosures, concluding that environmental disclosures by firms have no effect on the firm's market value and do not arbitrate the influence of FEP and FFP regarding firm value in Indonesia. Testa and D'Amato (2017) studied the bidirectional causal nexus between FFP and firm environmental responsibility regarding the listed manufacturing company data in Italy from 2005 to 2014. Meanwhile, their outcomes showed no support for the bidirectional hypothesis; however, firm environmental responsibility had a significant relationship with previous FFP, supporting the slack resources hypothesis. Giannarakis et al. (2017) studied the influence of FEP with respect to the environmental disclosure level of Standard and Poor's 500 firms, and their relevant results showed that higher pollution levels regarding greenhouse gas emissions had a negative effect on the distribution of carbon disclosure information, which suggests a positive nexus between a firm's environmental disclosure level and its FEP. In addition, Alipour et al. (2019) indicated that there was a significant positive nexus between FFP and environmental disclosure quality in Iranian firms.

Muhammad et al. (2015) found no connection between FEP and FFP from 2008 to 2010 (during the financial crisis) in Australia. Liang and Liu (2017) obtained the same result for Chinese firms. Based on a comprehensive meta-analysis of 149 stud-

ies, Endrikat et al. (2014) showed a positive and partly bidirectional nexus between FEP and FFP. Laguir et al. (2018) demonstrated a bidirectional nexus between FEP and FFP for French banks. Yu et al. (2009) made use of correlation analysis with data from of 51 European firms in 14 industries across 15 countries for the purpose of investigating the possible nexus between FFP and FEP; finding no positive nexus between FFP and FEP. Dragomir (2010) reported no nexus between FFP and FEP based on data from 60 of the biggest European Union industry commercial units. To date, the academic debate concerning the nexus between R&D investment, FEP ($CO_2$ emission) and FFP is ongoing.

## 3. Methodology

### 3.1. Empirical Model

Despite the causality specification of the Granger causality test (GCT, Granger 1969), which is unable to offer knowledge regarding nonlinear causalities or a tail causal relation, the quantile causality method analyzes causal nexuses in a more agile and careful way. In this study, the quantile regression Fourier Toda–Yamamoto (QRFT-Y) GCT proposed by Nazlioglu et al. (2016) is used to test the relations among R&D investment, $CO_2$ emissions and ROA in Taiwan. Via the QRFT-Y GCT, the subsequent Fourier series is used in lieu of utilizing dummy parameters to cogitate the structural breaks in the causal nexus. Gallant (1981) and Gallant and Souza (1991) indicate that a small number of low-frequency units for a Fourier function are able to catch an unidentified number of breaks regarding sharp as well as gradual structural breaks:

$$d(t) = \gamma_1 \sin\left(\frac{2\pi kt}{T}\right) + \gamma_2 \cos\left(\frac{2\pi kt}{T}\right) \tag{1}$$

This Fourier series, in which $k$ specifies the frequency, can be incorporated into the testing equation; its quantile approach is defined as follows:

$$\Delta Y_t(\tau) = c(\tau) + d(t) + \sum_{i=1}^{p-1} \theta(\tau)\Delta Y_{t-i} + \sum_{i=1}^{p-1} \delta(\tau)\Delta X_{t-i} + \sum_{i=1}^{p-1} r(\tau)\Delta Z_{t-i} + \varepsilon_t \tag{2}$$

or

$$\Delta Y_t(\tau) = c + \gamma_1 \sin\left(\frac{2\pi kt}{T}\right) + \gamma_2 \cos\left(\frac{2\pi kt}{T}\right) + \sum_{i=1}^{p-1} \theta(\tau)\Delta Y_{t-i} \\ + \sum_{i=1}^{p-1} \delta(\tau)\Delta X_{t-i} + \sum_{i=1}^{p-1} r(\tau)\Delta Z_{t-i} + \varepsilon_t \tag{3}$$

where $x$, $y$ and $z$ are R&D investment, $CO_2$ emissions and ROA, respectively. The GCT can be simulated from $z \rightarrow y$ at various quantiles regarding the null hypothesis of $\gamma(\tau_i) = 0$ for $i = 1, 2, \ldots, p$. Moreover, the null hypothesis without Granger causality, i.e., $R\beta(\tau) = 0$, is simulated as follows:

$$\text{Wald} = (R\beta(\tau))' \left[ R(Z'Z)^{-1} \otimes S \right] R')^{-1}](R\beta(\tau)) \tag{4}$$

where $R$ defines the indicator matrix of these variables (constrained variables denoted by ones), $\beta(\tau)$ denotes D′ column stack, $S$ represents the variance–covariance matrix for the unconstrained approach; $\otimes$ indicates the Kronecker product. Furthermore, considering that Hatemi-J and Uddin (2012) mention the fact that conditional heteroskedasticity (ARCH) influences exist in the data, and these do not usually abide by normal distribution; there is thus a probability that the Wald statistic distribution will significantly diverge from its asymptotic distribution. Hence, the bootstrapping simulation technique, using 10,000 iterations, proposed by Hatemi-J and Uddin (2012), is used to comprise the 1%, 5% and 10% critical values from the empirical distribution. Similarly, the Wald test can be utilized for testing of the null hypothesis: $\gamma(\tau_i) = 0$ for $i = 1, 2$.

*3.2. Data*

In this study, Taiwan Stock Exchange (TSE)-listed firms were used for sample data, obtained from the Taiwan Economic Journal (TEJ) database. Although the initial sample started with 2640 firm-year observations (542 firms) that were involved with $CO_2$ emissions, as stated in their financial reports from 2005 to 2018, up to 1097 of these (222 firms) were excluded, owing to 430 of them (90 firms) having missing accounting and financial data, and 667 (132 firms) having zero R&D investment, respectively. Thus, the final unbalanced panel data comprised 1543 firm-year observations (320 firms) from 2005 to 2018, covering the financial crisis (from 2007 to 2009) period.

These are cross-sectional and time-series data, permitting the overall examination of the trivariable nexus. This study's parameters were defined as follows. The $CO_2$ emissions were evaluated as the ratio of $CO_2$ emissions to assets in order to minimize heterogeneity problem which arises from different sizes. The return on assets (ROA) was evaluated as the ratio of net profit to assets. R&D was measured as the ratio of R&D investment to net sales revenue. Following Scarpellini et al. (2019), we used R&D investment as a metric for the measurement of innovation, and $CO_2$ emissions and ROA served as a proxy for FEP and FFP, respectively.

Table 1 presents the descriptive statistics for the unbalanced panel sample from 2005 to 2018. The total number of firms was 320, and there were 1543 firm-year observations. Jarque–Bera analysis showed that $CO_2$ emissions, ROA and R&D were non-normally distributed. The mean (median) value of $CO_2$ emission (divided by assets) was 0.1092 (0.0214), with a maximum value of 4.0409. The mean (median) value of ROA was 10.39072 (9.7600), with a maximum value of 55.64. The mean (median) value of R&D was 0.088068 (0.023), with a maximum value of 17.68196.

**Table 1.** Descriptive Statistics.

|  | Mean | Medium | Max. | Min. | Std.Dev. | Skewness | Kurtosis | Jarque–Bera |
|---|---|---|---|---|---|---|---|---|
| $CO_2$ | 0.1092702 | 0.021471 | 4.0409 | 0.0000031 | 0.33829 | 7.204648 | 64.13949 | 253,673.3 *** |
| ROA | 10.39072 | 9.760000 | 55.640 | −63.04000 | 8.286528 | −0.479822 | 10.49469 | 3670.491 *** |
| R&D | 0.088068 | 0.023018 | 17.68196 | $2.39 \times 10^{-5}$ | 0.778212 | 18.77636 | 375.0439 | 8,989,701 *** |

Notes: *** indicates significance at the 1% level. $CO_2$ indicates the ratio of carbon emission to assets, used in order to minimize the heterogeneity problem which arises from different sizes. ROA refers to the ratio of net profit to assets and R&D refers to the ratio of R&D investment to net sales revenue.

## 4. Empirical Results

Using a novel bootstrap Fourier quantiles Granger causality test to analyze data from 2005 to 2018, this study examines the nexus between R&D investment, FEP ($CO_2$ emissions) and FFP (ROA) in Taiwan. First, we carried out traditional unit root tests, followed by the test of bootstrap Fourier Granger causality in quantiles.

*4.1. Unit Root Test Results*

Numerous traditional unit root tests the ADF, PP and KPSS tests were applied and the outcomes are shown in Table 2. Only R&D was non-stationary, and this became stationary at level (I (0)) and after taking the first difference. $CO_2$ emissions and ROA were all stationary at level (I (0)) and after taking the first difference, respectively.

**Table 2.** Unit Root Test Results.

|  | Level | | | First Difference | | |
|---|---|---|---|---|---|---|
|  | **ADF** | **PP** | **KPSS** | **ADF** | **PP** | **KPSS** |
| $CO_2$ | −7.824175 *** | −11.63009 *** | 0.372556 | −12.97932 *** | −60.37167 *** | 0.079667 |
| ROA | −18.57293 *** | −23.46197 *** | 0.242253 | −23.46197 *** | −146.3223 *** | 0.077861 |
| R&D | −11.174 *** | −6.88788 *** | −50.16287 *** | −18.71048 | −50.16287 *** | 0.116063 |

Notes: *** indicates significance at the 1%, 5% and 10% levels, separately. $CO_2$, refers to the ratio of carbon emission to assets, used to minimize the heterogeneity problem which arises from different sizes. ROA refers to the ratio of net profit to assets and R&D refers to the ratio of R&D investment to net sales revenue.

### 4.2. Bootstrap Fourier Quantiles Granger Causality Test

To test the Granger causality, the existence of a Fourier expansion in equation (2) is tested using the conventional Wald test. The critical values are computed, using the bootstrapping procedure, with 10,000 replications. In Table 3, the BFQGCT results, running from R&D to $CO_2$ emissions, indicate that the values of the Wald statistics equal 4040.917, 1748.515, 14,956.70, 2048.546 and 1334.683 under 0.2, 0.3, 0.7, 0.8 and 0.9 quantiles, respectively, and each is greater than its bootstrap critical values at 5% or 1%. The results indicate that the null hypothesis of $\gamma\_1 = \gamma\_2 = 0$ can be strongly rejected at 5% or 1% levels of significance. There is a one-way Granger causality running from R&D investment to $CO_2$ emissions (under 0.2, 0.3, 0.7, 0.8 and 0.9 quantiles). By observing the symbol of the independent-variable coefficients, it can be seen that R&D investment causes $CO_2$ emissions to decrease. These findings demonstrate that R&D investment (innovation) reduces $CO_2$ emissions (increasing environmental performance) under 0.2, 0.3, 0.7 and 0.8 environment performance quantiles, respectively. The Granger causality running from $CO_2$ emissions to R&D is insignificant.

**Table 3.** Quantile Granger causality test among $CO_2$ emissions, ROA and R&D.

| Quantile | | Wald Test | CV 10% | CV 5% | CV 1% |
|---|---|---|---|---|---|
| | 0.1 | 2.137394 | 579.4584 | 1226.427 | 4204.114 |
| | 0.2 | 4048.917 ** | 581.3760 | 1164.403 | 4586.895 |
| | 0.3 | 1748.515 ** | 447.0656 | 853.5995 | 3288.686 |
| R&D→$CO_2$ | 0.4 | 11.50215 | 375.8133 | 645.3720 | 2315.774 |
| (negative) | 0.5 | 6.815864 | 393.0546 | 637.7543 | 1860.990 |
| | 0.6 | 3.146628 | 458.4620 | 752.7757 | 3216.239 |
| | 0.7 | 14,956.70 ** | 595.3574 | 1025.499 | 4906.720 |
| | 0.8 | 2048.546 ** | 553.0080 | 1133.514 | 5111.547 |
| | 0.9 | 1334.683 ** | 367.9836 | 814.6717 | 3258.381 |
| | 0.1 | 3.880787 | 89.13272 | 125.0902 | 263.7421 |
| | 0.2 | 1.951739 | 36.83624 | 50.19736 | 87.53883 |
| | 0.3 | 1.069226 | 23.23389 | 30.37297 | 47.74848 |
| | 0.4 | 3.064328 | 20.97151 | 26.13679 | 41.64357 |
| $CO_2$→R&D | 0.5 | 7.655415 | 22.86247 | 28.58717 | 44.25323 |
| | 0.6 | 9.659755 | 29.81040 | 36.59088 | 57.44164 |
| | 0.7 | 2.334995 | 41.04957 | 50.95854 | 76.42753 |
| | 0.8 | 0.898962 | 65.85857 | 84.46474 | 136.1751 |
| | 0.9 | 3.004358 | 108.4133 | 147.1132 | 268.6119 |
| | 0.1 | 15.70273 | 65.06867 | 85.70210 | 145.4731 |
| | 0.2 | 2.602070 | 25.97153 | 35.32248 | 61.96993 |
| | 0.3 | 8.946593 | 15.69944 | 21.71965 | 38.12811 |
| $CO_2$→ROA | 0.4 | 7.295380 | 12.15180 | 16.66797 | 29.97618 |
| (negative) | 0.5 | 17.20173 ** | 12.27122 | 16.51336 | 28.73321 |
| | 0.6 | 27.32757 ** | 14.92517 | 19.52453 | 32.64043 |
| | 0.7 | 19.32699 | 21.67174 | 28.28871 | 46.05749 |
| | 0.8 | 57.70779 ** | 37.64852 | 48.66073 | 87.00708 |
| | 0.9 | 119.3797 ** | 77.38182 | 100.8456 | 174.9821 |
| | 0.1 | 0.532327 | 23.39925 | 27.08539 | 34.32783 |
| | 0.2 | 3.043201 | 16.08705 | 18.15568 | 22.78126 |
| | 0.3 | 3.958516 | 15.65950 | 17.73498 | 21.70123 |
| | 0.4 | 3.158619 | 15.99712 | 18.09925 | 22.31895 |
| ROA→$CO_2$ | 0.5 | 2.501898 | 15.82778 | 17.92695 | 22.32631 |
| | 0.6 | 3.426661 | 16.38645 | 18.86070 | 24.30664 |
| | 0.7 | 6.121450 | 19.46645 | 22.38043 | 29.09297 |
| | 0.8 | 7.133838 | 27.26242 | 32.11197 | 41.93813 |
| | 0.9 | 32.68547 | 52.58201 | 64.05667 | 87.97280 |

**Table 3.** *Cont.*

| Quantile | | Wald Test | CV 10% | CV 5% | CV 1% |
|---|---|---|---|---|---|
| R&D→ROA (positive) | 0.1 | 341.3374 | 575.3449 | 663.1935 | 938.3257 |
| | 0.2 | 386.0742 | 445.2968 | 524.5006 | 754.4572 |
| | 0.3 | 411.8341 * | 353.6841 | 422.8852 | 578.4530 |
| | 0.4 | 397.0857 ** | 335.1044 | 386.8160 | 514.1024 |
| | 0.5 | 470.3310 ** | 322.8919 | 375.7900 | 486.7262 |
| | 0.6 | 148.1905 | 270.1801 | 340.4037 | 495.5135 |
| | 0.7 | 128.7050 | 379.8922 | 467.0425 | 679.6993 |
| | 0.8 | 370.8866 | 567.1032 | 672.3535 | 943.0788 |
| | 0.9 | 1289.359 ** | 849.3702 | 1020.328 | 1404.347 |
| ROA→R&D (negative) | 0.1 | 3.043362 | 37.08690 | 44.13056 | 59.42251 |
| | 0.2 | 9.743216 | 22.38750 | 25.82377 | 33.62489 |
| | 0.3 | 15.85482 | 20.22187 | 23.12610 | 28.80851 |
| | 0.4 | 18.23258 | 18.48369 | 20.93514 | 26.67804 |
| | 0.5 | 35.66072 *** | 17.96496 | 20.67274 | 26.01630 |
| | 0.6 | 22.96598 ** | 18.95430 | 21.75916 | 27.86017 |
| | 0.7 | 7.064264 | 19.63218 | 22.91689 | 30.25602 |
| | 0.8 | 16.28614 | 25.96321 | 30.44414 | 41.30809 |
| | 0.9 | 24.86138 | 56.45294 | 66.90733 | 90.15367 |

Notes: The frequency of the data is monthly. CV10%, CV 5%, and CV1% are the critical values of statistical significance, individually. *, ** and *** indicate rejection at the 10%, 5% and 1% levels, individually.

Table 3 shows the results of the Granger causality test running from $CO_2$ emissions to ROA, indicating that the values of the Wald statistics equal 17.20173, 27.32757, 57.70779 and 119.3797 under 0.5, 0.6, 0.8 and 0.9 quantiles, respectively, and each is greater than its bootstrap critical values at 5%. The results indicate that the null hypothesis of $\gamma\_1 = \gamma\_2 = 0$ can be strongly rejected at the 5% level of significance. There is a one-way Granger causality running from $CO_2$ emissions (environmental performance) to ROA (financial performance) (under 0.5, 0.6, 0.8 and 0.9 quantiles). The symbol of the independent-variable coefficients indicates that $CO_2$ emissions cause ROA to decrease. This means that reducing emissions (increasing environmental performance) "Granger-causes" an increase financial performance under 0.5, 0.6 0.8 and 0.9 financial performance quantiles, respectively. The Granger causality running from ROA to $CO_2$ emissions is insignificant.

The Granger causality test running from R&D to ROA indicates that the values of the Wald statistics equal 411.8341, 397.0857, 470.3310 and 11,289.359 under 0.3, 0.4, 0.5, and 0.9, respectively, and each is greater than its bootstrap critical values at 10% or 5%. The outcomes show that the null hypothesis of $\gamma\_1 = \gamma\_2 = 0$ can be strongly rejected at the 10% or 5% level of significance. There is a one-way Granger causality running from R&D to ROA (under 0.3, 0.40, 0.5 and 0.9). By examining the symbol of the independent-variable coefficients, it can be seen that increasing R&D causes ROA to increase. These findings conclude that increasing R&D investment increases FFP under 0.3, 0.4, 0.5 and 0.9 financial performance quantiles, respectively.

The Granger causality test running from ROA to R&D indicates that the values of the Wald statistics equal 35.66072 and 22.96598 under 0.5 and 0.6, respectively, and each is greater than its bootstrap critical values at 1% and 5%. The outcomes show that the null hypothesis of $\gamma\_1 = \gamma\_2 = 0$ can be strongly rejected at the 1% and 5% level of significance. The Granger causality runs from ROA to R&D (under 0.5 and 0.6 quantiles). By examining the symbol of the independent-variable coefficients, it can be seen that increasing ROA causes R&D investment to decrease, i.e., the better the FFP, the lower the R&D investment under the middle R&D investment quantiles. The causality nexus, as well as the pathways among R&D investment, $CO_2$ emission and financial performance, are shown in Figure 1.

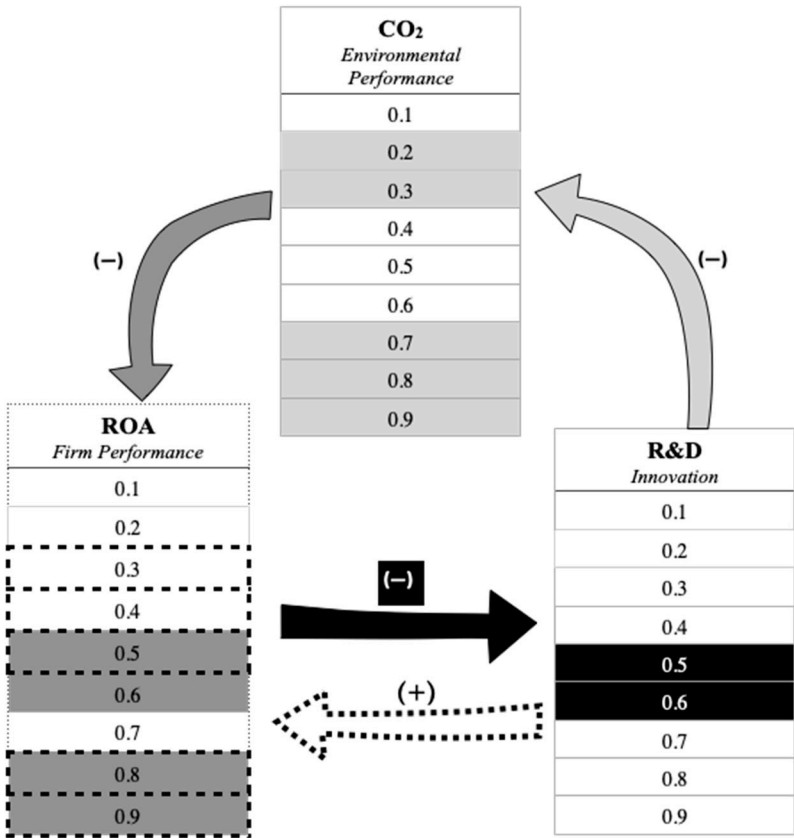

**Figure 1.** Quantile Granger Causality Test Direction.

## 5. Conclusions

As CO$_2$ emissions affect global warming, social concern about the environment is increasing. These increasing social concerns, along with the question of whether "it pays to be green" in business, have motivated firms to seek solutions in regard to decreasing CO$_2$ emissions and increasing profitability simultaneously. In this study, we have examined the nexus between R&D investment, FEP (CO$_2$ emissions) and FFP in Taiwan via a novel BFQGCT using data from 2005 to 2018. The empirical outcomes show that R&D investment causes an improvement in both CO$_2$ emissions and ROA. By examining the results of the independent-variable coefficients, it can be observed that R&D investment (innovation) reduces CO$_2$ emissions (boosting FEP) under 0.2, 0.3, 0.7, 0.8 and 0.9 FEP quantiles, separately, and R&D investment increases ROA (FFP) under 0.3, 0.4, 0.5 and 0.9 FFP quantiles, individually. Reducing CO$_2$ emissions (FEP) causes an increase in ROA (FFP) under 0.5, 0.6, 0.8 and 0.9 ROA quantiles, separately, and increasing ROA (FFP) causes reduced R&D investment under 0.5 and 0.6 quantiles, individually. Finally, increasing R&D causes an increase in ROA under 0.3, 0.40, 0.5 and 0.9 ROA quantiles, individually, in Taiwan.

Contrary to the findings of a previous study that used only mean coefficient analysis via the OLS method, the analyses presented here indicate that the greater the R&D investment, the greater the reduction in CO$_2$ emissions under 0.2, 0.3, 0.7, 0.8 and 0.9 environment performance quantiles, individually. In contrast with previous findings (Scarpellini et al. 2019; Duque-Grisales et al. 2020; Ezhilarasi and Kabra 2020), our results show that a larger R&D investment is linked with better financial performance under 0.3, 0.4, 0.5 and 0.9 quantiles, separately. The negative causality running from ROA to R&D under 0.5 and 0.6 R&D investment quantiles, individually, indicates that better financial performance is linked with the less R&D investment in middle R&D investment quantiles. These findings suggest that there is a negative causality running from environmental performance to financial performance under the 0.5, 0.6, 0.8 and 0.9 financial performance quantiles, individually, which offers a potential explanation for the inconclusive findings

in the previous literature (Liu et al. 2017; Moneva and Ortas 2010; Clarkson et al. 2011; Gallego-Álvarez et al. 2015; Muhammad et al. 2015). Understanding this causal nexus, as well as the pathways among R&D investment, $CO_2$ emission and financial performance, has special significance for policy makers intending to understand how R&D strategies can improve FEP and FFP simultaneously. This study is unique among prior studies in that it examines the direction and size of R&D investment, financial and environmental performance relations via a bootstrap Fourier quantiles Granger causality test. Due to the limitations of this model, it was not possible to analyze how the quantiles of the explanatory variables affected the conditional quantiles of the dependent variables. Further studies can hopefully make up for this limitation.

**Author Contributions:** F.-L.L. conceptualized the idea of this paper, collected and analyzed the data. F.-L.L. also wrote the manuscript. The author has read and agreed to the published version of the manuscript.

**Funding:** This research received no external funding.

**Institutional Review Board Statement:** Not applicable.

**Informed Consent Statement:** Not applicable.

**Data Availability Statement:** The datasets generated for this study are available on request to the author.

**Conflicts of Interest:** The author declares no conflict of interest.

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
