# Peer review of "R&D Investment, Financial and Environmental Performance Nexuses via Bootstrap Fourier Quantiles Granger Causality Test"

_economies, doi:10.3390/economies9020085_

Round 1
Reviewer 1 Report
Hello Author,
The article presented interesting research results. The model and the explanation of the results are strong elements of the article.
The following areas need improvement:
1) Rewrite the abstract of the article for clarity. Break down long sentences into short sentences to make them more understandable.
2) The data and the descriptive statistics need to be presented in detail to inform the readers of the nature of the data used in the research.
3) Granger Causality Test has limitations. The limitations should be indicated in the article. The results should also be tested with an alternative model to give more credibility to the research results.
Reviewer 2 Report
Your study is interesting and relevant to pressing problems with environmental degradation in the face of economic development.
Your assertions are well supported and your methodology is rigorous and clearly explained. The research question is properly motivated and the literature review is expansive. Nice work!
My primary suggestion is to have the document carefully proofread by someone for whom English is their first language. The writing currently suffers from grammatical errors and there are problems with spacing and syntax as well. In it's present form, the writing issues would distract from the importance of your findings, so please do have someone check the writing in your manuscript.
Overall, this is a valuable contribution to the literature.
Round 2
Reviewer 1 Report
Dear Authors,
The responses to my comments address some of the issues. However, you need to address the issues I raised in detail. The abstract still needs editing.
Round 3
Reviewer 1 Report
The paper still needs editing before publication, including rewriting the abstract.
